



# Determining the dependence of the power supply to the ocean on the length and time scales of the dynamics between the meso-scale and the synoptic-scale, from satellite data

Achim Wirth[1]

[1]Univ. Grenoble Alpes, CNRS, Grenoble INP, LEGI, 38000 Grenoble, France

**Correspondence:** achim.wirth@legi.cnrs.fr

**Abstract.** The input of mechanical power to the ocean due to the surface wind-stress, in regions which correspond to different regimes of ocean dynamics, is considered using data from satellites observations. Its dependence on the coarse-graining range of the atmospheric and oceanic velocity in space from $0.5^o$ to $10^o$ and time from 6h to 40 days is determined. In the area of the Gulf Stream and the Kuroshio extensions the dependence of the power-input on space-time coarse-graining varies over tenfold

for the coarse-graining considered. It decreases over twofold for the Gulf Stream extension and threefold for the Kuroshio extension, when the coarse-graining length-scale passes from a few degrees to $0.5^o$ at a temporal coarse-graining scale of a few days. It increases over threefold in the Gulf Stream and the Kuroshio extensions when the coarse-graining passes from several days to 6h at a spatial coarse graining of a few degrees. The power input is found to increase monotonically with shorter coarse-graining in time. Its variation with coarse graining in space has no definite sign. Results show that including the dynamics at

scales below a few degrees reduces considerably the power input by air-sea interaction in regions of strongly non-linear ocean currents. When the ocean velocities are not considered in the shear calculation the power-input is considerably (up to threefold) increased. The dependence of the power input on coarse graining in space and time is close to being multiplicatively separable in all regions and for most of the coarse-graining domain considered.

## 1 Introduction

The mechanical power supply to the ocean is mainly due to the friction of the atmospheric winds at the ocean surface. It involves a multitude of processes ranging from molecular dynamics as evaporation and droplet formation (Veron (2015)) to wave generation, interaction and breaking (Melville (1996)), to generation of coupled synoptic-scale instabilities (Moulin and Wirth (2014)), up to the basin scale modes (Bjerknes (1964)) of variability, as El Ninio (Alexander et al. (2002)) or the North Atlantic oscillation (Hurrell et al. (2003)). An abundance, in type and occurrence, of ocean waves and currents transport the

power from the location where it is injected to different locations where it does work or is dissipated, and also storing it on a large range of time-and-space scales as kinetic or potential energy. The synoptic scale of the wind, that is the scale of the weather systems ($\approx 1000$km) at mid-latitude, (see e.g. Vallis (2017)) is usually an order of magnitude larger than the corresponding dynamical-scale in the ocean, the scale of the oceanic eddies, called the meso-scale. The typical associated time scale is a few days for the atmosphere and a few weeks for the ocean eddies. Only a small part of the mechanical power





injected in the ocean by the atmosphere leads to coherent ocean currents and eddies over larger horizontal ($> 100$km), vertical ($> 100$m) and temporal ($> 1$week) scales (see e.g. Ferrari and Wunsch (2009)). The supply of mechanical power due to the dynamics between the meso-scale and the synoptic-scale and its spatio-temporal structure is the subject of the present work.

The classical picture concerning the ocean circulation states that the ocean is forced at the basin scale by the large-scale

atmospheric winds, which vary on inter-annual and seasonal time scales. The abundant energy at the eddy scale ($\approx 100$km) in the ocean is found to be mostly supplied by baroclinic and barotropic instability of the basin scale circulation and the extensions of the western boundary currents to the ocean interior. The energy is further distributed across scales by the non-linear interactions, possibly leading to energy cascades over extended ranges of scales. This picture is still valid to a large extent, but has been refined recently (see e.g. Vallis (2017)). Before studying the energy cycle within the ocean, it is, however,

key to determine which scales of the dynamics are important for the supply and the draining of energy at the ocean surface and have to be included whenever the energy balance of the ocean dynamics is considered. This is the purpose of this publication. There is a subtle difference between the scale at which the power is fed to the ocean dynamics, the scale of the forcing by the wind-stress and the importance of the dynamics of the atmosphere and the ocean at a certain scale for the power-input, due to the nonlinear dependence of the stress and the power-input on the ocean currents and the atmospheric winds. More precisely,

the wind-stress vector of a turbulent flow over a surface is commonly expressed by:

$$\tau = C_D \rho_a \sqrt{(\mathbf{u_a} - \lambda \mathbf{u_o})^2}(\mathbf{u_a} - \lambda \mathbf{u_o}), \tag{1}$$

where $\rho_a$ is the density of the atmosphere, $\mathbf{u_a}$ the horizontal wind velocity vector near the surface, usually taken 10m above the surface and $\mathbf{u_o}$ the ocean current vector near the surface. The drag coefficient $C_D$ depends on the temperature stratification in the atmosphere and the roughness of the ocean surface, these phenomena will not be considered here and we refer the reader

to the classical books by Gersten et al. (2017) for boundary-layer theory , Stull (2012) for the planetary boundary layer and Csanady (2001) for air-sea interaction. For a calculation of the shear respecting Newton's laws $\lambda = 1$, for simplicity $\lambda = 0$ was often used in the past, especially when atmosphere-only or ocean-only models were numerically integrated (see Duhaut and Straub (2006) for a discussion) but also when data was analysed (see Renault et al. (2017) for a discussion).

The focus of previous research on mechanical power input to the ocean was on the temporal variability of the forcing and

the global magnitude and patterns of power input, as in Scott and Xu (2009), Roquet et al. (2011), Zhai et al. (2012) and Rimac et al. (2013). In the present work I focus on the contribution of the atmosphere and ocean dynamics at a given scale in space and time to the injection of power. The mechanical power injected:

$$\tilde{\mathcal{P}} = \tau \cdot \tilde{\mathbf{u}}_\mathbf{o}, \tag{2}$$

is given by the vector product of the wind-stress and the current, where the horizontal ocean current $\tilde{\mathbf{u}}_\mathbf{o}$, is not necessarily at the

surface. When the power supply to the geostrophic large scale circulation is considered, as in Stern (1975), Roquet et al. (2011) and Scott and Xu (2009), it is the geostrophic current near the ocean surface, usually obtained through satellite measurements of the sea-surface-height. In the present work I focus on the power provided to the mixed layer and $\tilde{\mathbf{u}}_\mathbf{o}$ is the horizontal ocean current at 15m depth. Straight forward calculations on the stationary Ekman-spiral show, that the difference of power injected to the surface current and the current below the Ekman-spiral is equal to the power locally dissipated by internal friction in the





Ekman layer. Using a $1/10^o$ global ocean model Rimac et al. (2016) estimated that less than one third of the power injected into the ocean at the surface by wind shear is transmitted below the mixed layer. Equations (1) and (2) are products and the scale (space-and-time) of the dynamics is not the scale of the wind stress, which is not the scale of the power-input. If, for example, the forcing is at the very large scale the power enters (or leaves) the flow at the scale close to the ocean current and not at the

scale of the forcing. The interesting property of eq. (2) is that the first factor is dominated by the atmosphere and the second given by the ocean dynamics, which usually operate at different scales in space and time, as stated above. The occurrence of both factors makes the power-input an interesting phenomena when scales, in-time-and-space are considered. This has to the best of my knowledge not been done before.

Recently the effect of "eddy killing" is subject of increased interest, as traditionally the ocean velocity was not included

in the shear calculation, meaning that $\lambda = 0$ in eq. (1) (Duhaut and Straub (2006), Renault et al. (2017)). Eddy-killing is the reduction of power input when the correct value $\lambda = 1$ is used instead of $\lambda = 0$. A justification of neglecting the ocean velocity is that near the surface the atmospheric winds are typically over an order of magnitude larger than the ocean currents and so the former can be neglected with respect to the latter in the shear calculation. However, when the power input is calculated, eq. (2) shows that the scalar product of the wind vector $\mathbf{u_a}$ and the current vector $\tilde{\mathbf{u}}_\mathbf{o}$ are compared to the product of the surface current

$\mathbf{u_o}$ and the current vector $\tilde{\mathbf{u}}_\mathbf{o}$. To evaluate the importance not only the magnitudes of the vectors have to be considered, but also their alignment. The correlation of the ocean current considered and the ocean surface current leads, typically to a reduction of the power input, a phenomena that is often referred to as "eddy killing". This was observed in numerical simulations (see i.e. Duhaut and Straub (2006)) and satellite observations (see i.e. Renault et al. (2017)) and it was analytically calculated for a highly idealised model in Wirth (2018). In recent publications the effect of eddy-killing was quantified to compensate for it by

modifying the value of the drag coefficient. This is questionable as eddy killing operates differently on different scales in space and time and for different dynamical regimes. Eddy-killing is not restricted to eddies, its quantification over a large range of scales in space and time is considered here.

I consider two types of dynamical regimes: the extension of western boundary currents and the re-circulation area in the low-latitude part of the subtropical gyre. To evaluate the significance of the results the two types of regions are considered in

the North Atlantic and the North Pacific.

## 2  Theory

We consider the normalised power $\mathcal{P} = \tilde{\mathcal{P}}/(\rho_a C_D)$, that is the power divided by the density of air and the drag coefficient. The drag coefficient depends not only on physical parameters but also on the scale in space and time of the data or the resolution of the model and this dependence is not known. A better determination of the drag coefficient is one of the purposes of the present

research. Values of $\mathcal{P} = 1(\text{m/s})^3$ correspond to roughly $\tilde{\mathcal{P}} \approx 2 \cdot 10^{-3}\text{W/m}^2$. The velocities in the atmosphere and the ocean are averaged in time and space. In the present work I determine the scales which are important for the mechanical power supply to the ocean. To this end a velocity vector is first averaged over a horizontal square of length $l$ and a time interval $\tau$ to obtain the

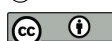



coarse-grained velocity:

$$\overline{\mathbf{u}}^{l,\tau}(x,y,t) = \frac{1}{l^2\tau} \int\limits_{-\tau/2}^{\tau/2} \int\limits_{-l/2}^{l/2} \int\limits_{-l/2}^{l/2} \mathbf{u}(x+x',y+y',t+t')dx'dy'dt'. \tag{3}$$

The normalised power is then calculated based on the averaged quantities and averaged over the entire domain and observation interval:

$$\mathcal{P}(l,\tau;\lambda) = \frac{1}{(L_x-l)(L_y-l)(t_e-t_o-\tau)} \int\limits_{l/2}^{L_x-l/2} \int\limits_{l/2}^{L_y-l/2}$$

$$\int\limits_{t_0+\tau/2}^{t_e-\tau/2} \sqrt{(\overline{\mathbf{u}}_a^{l,\tau}-\lambda\overline{\mathbf{u}}_o^{l,\tau})^2} \ (\overline{\mathbf{u}}_a^{l,\tau}-\lambda\overline{\mathbf{u}}_o^{l,\tau}) \cdot \overline{\tilde{\mathbf{u}}}_o^{l,\tau} \, dt \, dx \, dy. \tag{4}$$

It varies with averaging length $l$ and time interval $\tau$ due to the nonlinear dependence of the power on the velocities. If a model or an observation has a resolution $l_0$ it follows that $\mathcal{P}(l,\tau) = \mathcal{P}(l_0,\tau) \, \forall \, l \leq l_0$. The same applies to coarse-graining in time. The power input obtained from the coarse grained variables tells us to what extent the dynamics at a certain scale in space and time contributes to the power input.

To quantify the effect of eddy-killing we consider:

$$r_{\text{kill}}(l,\tau) = (\mathcal{P}(l,\tau;0) - \mathcal{P}(l,\tau;1))/\mathcal{P}(l,\tau;1). \tag{5}$$

Including the ocean velocity in the shear calculation usually reduces the power supply and $r_{\text{kill}} > 0$.

I further consider:

$$\mathcal{P}_m(l,\tau) = \frac{\mathcal{P}(l,10\text{days})\mathcal{P}(5^o,\tau)}{\mathcal{P}(5^o,10\text{days})}, \tag{6}$$

if it equals $\mathcal{P}(l,\tau)$ the latter is said to be multiplicatively separable and the dependence on time scale and length scale can be considered independently. In this case the relative importance of a certain length-scale for the power injected is independent of the corresponding time-scale and vice-versa.

## 3 Data

Data for ocean currents and wind is provided through the Copernicus Marine Environment Monitoring Service (CMEMS, http://marine.copernicus.eu/services-portfolio). More specifically for the ocean current data the GlobCurrent product is used which is based on the estimation of ocean surface currents from satellite sensor synergy through the combination of altimetry, GOCE, wind and in-situ data (Rio et al. (2014)). For a critical discussion of the data and its limitations we refer to Bonjean and Lagerloef (2002) and Sudre et al. (2013). This global ocean surface current product covers consistently the last 25 years.





The winds are estimated from scatterometer and radiometer wind observations based on several missions (ERS-1, ERS-2, QuikSCAT, and ASCAT) available at $1/4^o$ spatial resolution and every 6 hours. For a critical discussion of the data and its limitations we refer to Bentamy et al. (2017) and Desbiolles et al. (2017). The data record for which both wind and current data is available spans 24 years, 1993–2016, at a resolution of 6h in time and $1/4^o$ in space.

Four domains, each spanning $10^o$ in the latitudinal and longitudinal directions, are considered. The first is in the subtropical gyre ($20^o-30^oN, 20^o-30^oW$) of the North Atlantic (ASG), the second in the Gulf Stream extension ($35^o-45^oN, 35^o-45^oW$) (GSE), the third is in the subtropical gyre of the North Pacific ($15^o-25^oN, 150^o-160^oE$) (PSG) and the fourth in the Kuroshio extension ($30^o-40^oN, 150^o-160^oE$) (KUE). At four occasions in time data was missing, the gaps were filled by linear interpolation. There are no land points, islands or coast-lines, in the regions. The power input is considered for spatial coarse

graining $l \in [0.5^o, 10^o]$ at a resolution of $0.5^o$ and temporal coarse graining $\tau \in [6h, 40\,\text{days}]$ at a resolution of 6h for $\tau \in [6h, 1\,\text{day}]$, 1 day for $\tau \in [1\,\text{day}, 10\text{days}]$ and 4 days for $\tau \in [10\,\text{days}, 40\,\text{days}]$.

## 4   Results

The normalised power-input $\mathcal{P}(l, \tau, 1)$ as a function of the coarse-graining scales in space $l$ and time $\tau$ is plotted in figure 1, for the four regions. In the re-circulation of the subtropical gyres (ASG and PSG) the values of the power input vary by a factor

of two and three, while the variation exceeds ten for the extensions of western boundary currents (GSE & KUE), for the range of coarse-graining scales considered. The power input is found to increase monotonically with shorter coarse-graining in time but not with finer coarse-graining in space (see figs. 1 & 2). The power-input increases over threefold in the GSE and the KUE when the coarse-graining passes from several days to 6h at a spatial coarse-graining of a few degrees (see fig. 2). For all regions the overall minimum of power-input is located at the longest coarse-graining in time (40 days). While for the regions within

the subtropical gyres the minimum is at a few degrees in space, it is at the smallest spatial coarse-graining scale ($0.5^o$) for the extensions of the western boundary currents. For scales larger than a few degrees and a few days, the variability of the power input with the coarse graining is less pronounced. Variations for $\tau > 10$ days and $l > 5^0$ are less than 25% for ASG, PSG while they attain almost unity for GSE and 2/3 for KUE.

    The dependence of the power input on coarse graining in space and time is close to being multiplicatively separable in

all regions and for most of the coarse-graining domain considered (see eq. (6) and fig. 1). The multiplicatively separable approximation $\mathcal{P}_m(l, \tau)$ agrees with the data along the lines $l = 5^o$ and $\tau = 10$ days, by definition (see eq. 6) and is small for the majority of coarse graining considered, the contour line for the 5% error is shown as a white line in fig.1. The major part of the coarse graining domain is within the 5% error contour line and higher errors are found only in the corners. For ASG the relative error is always below 5%, for PSG it is below 10%, for KUE it is below 20% and for GSE its maximum error is

almost 23%. In the GSE and the KUE the extremes of the relative difference $(\mathcal{P}_m - \mathcal{P})/\mathcal{P}$ are at $\tau = 6h$, with a maximal under-estimation by $\mathcal{P}_m$ for the finest coarse-graining scale $l = 0.5^o$ and a maximal over-estimation for the coarsest coarse-graining scale $l = 10^o$.





**Figure 1.** Contour plot of normalised power-input $\mathcal{P}(l, \tau, \lambda = 1)$ with isolines (in black), same isolines for $\mathcal{P}(l, \tau, \lambda = 0)$ are superposed in blue. Upper-left: ASG (isolines: 1, 1.25, 1.5), upper-right: GSE (isolines: 1, 2, 3, 4), lower-left: PSG (isolines: 1, 1.25, 1.5, 1.75), lower-right: KUE (isolines: 1 (blue line not present), 2, 3, 4). The white lines give the relative error at $\pm 5\%$ of the multiplicatively separable version $(\mathcal{P}_m - \mathcal{P})/\mathcal{P}$ (see eq. (6)).

We further consider the eddy-killing through the parameter $r_{\text{kill}}(l, \tau)$ defined in eq. (5). It is positive for all averaging scales and time-intervals $(l, \tau)$ (not shown), that is "eddy killing" always reduces the power input (fig. 2 and also a comparison of the




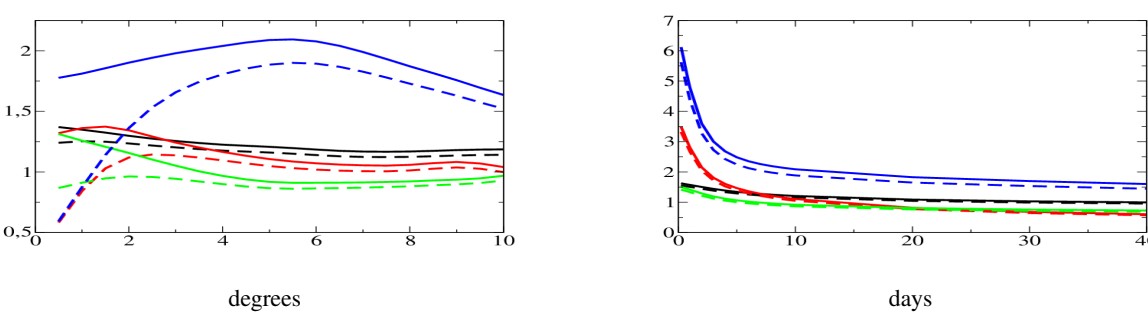

degrees

days

**Figure 2.** Normalised power-input $\mathcal{P}(l,\tau,\lambda)$; at $\tau = 10$ days (left), $l = 5^o$ (right); for $\lambda = 0$ (full-line), $\lambda = 1$ (dashed-line); for ASG (black), GSE (red), PSG (green) and KUE (blue).

blue isolines and black isolines in fig. 1). This comes at no surprise as it says that the ocean velocity at the surface and at 15m depth are on average positively correlated. Eddy killing becomes stronger for smaller scales in all cases and for all averaging times $\tau$. It is maximum at the smallest scale considered $l = 0.5^o$ and for $\tau \approx 8$ days in ASG, PSG and KUE and $\tau \approx 20$ days for GSE. In all cases considered $r_{\text{kill}}(l,\tau) < 10\%$ for $l > 4^o$, but exceeds $r_{\text{kill}}(l,\tau) > 10\%$ for $l < 2^o$ in all cases except ASG

where maximal values are slightly smaller than $10\%$. In all cases eddy killing depends strongly on length-scale averaging and weakly on time-scale averaging. It exceeds two for $l = 0.5^o$ in KUE, which corresponds to a three-fold over-estimation of the power-input, when eddy-killing is neglected ($\lambda$ put to zero). An "eddy-killing", that is a decrease of the energy input at small scale appears also in the cases with $\lambda = 0$, (see blue contour-lines fig. 1 and also full lines fig. 2) it is however much less pronounced. The source of it may lie in the data of the wind velocity which might incorporate a degree of shear velocity, that

is ocean dynamics.

In the multiplicatively separable approximation eddy killing is overestimated for the shortest and largest averaging times When the eddy-killing effect can be parameterised by changing the $C_D$ coefficient $r_{\text{kill}}(l,\tau)$ should be close to constant, which is roughly the case for $l > 5^o$ (see fig. 1 and also fig. 2).

## 5 Discussion

Results are strikingly different between the four different regions and the differences clearly put a constraint on empirically obtained parameterisation of small-scale dynamics in time and space. But similarities exist, especially in the regions with common dynamical regimes. The extension of the western boundary currents show a strong eddy killing at small scales. In the recirculation area of the gyres the minimal power input is for the longest averaging time and a spatial averaging between $5$ to $10^o$.

Using data from observations Zhai et al. (2012) found a roughly $70\%$ increase of the power input to the surface geostrophic circulation, when 6-hourly winds are used instead of monthly winds, for the global ocean. The power input further increases



when even higher forcing frequencies are considered. This high frequency forcing might not be important for the geostrophic circulation but might be important only for the mixed layer, as noted by Zhai et al. (2012). this is why I considered the power-input to the mixed layer, rather than the geostrophic circulation, in the present work. In global ocean circulation model Rimac et al. (2013) showed an almost fourfold increase in the near inertial power when increasing the resolution in time and space

from 6-hourly and $1.875^o$ to 1-hourly and $0.35^o$. They also note that the increase is dominated by the temporal scales over spatial scales due to the resonance condition at the inertial frequency.

In the present work I consider the dependence on resolution in time and space for limited regions of the world ocean. I found a 2.75 times increase of the power-input in the extensions of the western boundary currents and a 30 to $50\%$ increases in the subtropical gyre when the averaging period was reduced from one week to 6 hours at a spatial resolution of $5^o$ (see fig 2). It

is interesting to note that the strong decline in the power-input, in the extensions of the western boundary currents, between a day and a week is well fitted by a $\mathcal{P}(5^o, \tau) \propto \tau^{-1/2}$ law (not shown). The increase of energy input at 10 days from $0.5^o$ to $4^o$ (see fig 2) is well fitted by a $\mathcal{P}(l, 10 \text{ days}) \propto l^{1/2}$.

The eddy killing is found to be especially strong in the extension of the western boundary currents, where the energy input can drop by a factor between two and three when the averaging length scale is reduced from $5^o$ to $0.5^o$. The decay does not stop

or saturate at the smallest scale considered. It means that air sea-interaction due to the sub-meso-scale dynamics substantially decreases the power-input. The power input is maximum for a coarse-graining of $6^o$ for KUE and around $3^o$ for GSE. Meaning that the dynamics at smaller scales reduces the power-input.

## 6 Conclusions

The strongest gradients of power input are found for the smallest and shortest coarse-graining scales, that is within the coarse-

graining domain considered here we do not see any convergence of the power input. The problem is further complicated by the opposing gradients in space (including smaller scales reduces the power input) and space (including shorter scales increases the power input). This indicates that higher resolutions in space and time are necessary to estimate the power fluxes into the ocean mixed layer and that we do not know at which scales we possibly observe convergence. Furthermore, we do not know to what extent the power input due to the dynamics at a certain scale in time and space contributes to climate relevant processes

and how they can be parameterised. Our results also says that up- or down-scaling of data beyond the coarse-graining domain explored is not possible. A direct consequence of the strong variability is that the drag coefficient has to be a function of the coarse-graining scale when analysing observations and when performing numerical simulations of ocean dynamics. This is the reason why we considered the normalised power input, independent of the drag coefficient. The result, that the power input as a function of coarse-graining is close to multiplicatively separable reduces considerably the complexity of the problem, as the

space and time domain can be considered separately.

Diagrams of $\mathcal{P}(l, \tau)$ allow to estimate the change in the power injection when an observation or a numerical model is refined in space or higher frequency winds are considered for the forcing. In numerical simulations the cost of calculation of increasing the forcing frequency is almost negligible. Increasing the horizontal resolution of a numerical model is expensive as also the

number of vertical levels has to be increased to resolve the corresponding vertical modes and the time step has to be decreased to respect the CFL condition.

We have here considered the question about the contribution to the power input of the dynamics at a certain scale in space and time to the the mixed layer dynamics. In agreement with Zhai et al. (2012) and may other recent studies, this work emphasises

5   the importance of high-resolution coupled ocean-atmosphere models using a hierarchy of models and approaches.

*Data availability.* Copernicus Marine Environment Monitoring Service (CMEMS, http://marine.copernicus.eu/services-portfolio)

*Author contributions.* all research and writing performed by AW

*Competing interests.* None

*Acknowledgements.* This work was funded by Labex OASUG@2020 (Investissement d'avenir - ANR10 LABX56). These data were pro-

10  vided by the Centre de Recherche et d Exploitation Satellitaire (CERSAT), at IFREMER, Plouzane (France) and CMEMS. Part of this work was performed when AW visited LOPS, Brest. We are grateful to Abderrahim Bentamy for explanation concerning the data and Mickael Accensi and Jean-Fancois Piolle for help with the data analysis.



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
