# Peer review of "Determining the dependence of the power supply to the ocean on the length and time scales of the dynamics between the meso-scale and the synoptic-scale, from satellite data"

_Ocean Science, 2019_

## Author Comment (AC1) · 17 Jan 2020

In the discussion section, page 8 line 21 should read:

opposing gradients in space (including smaller scales reduces the power input) and time (including shorter scales increases

---

## Referee Comment (RC1) · Anonymous Referee #1 · 19 Apr 2020

This study aims to explore the dependency of wind energy input to the ocean on the scales of time and space. This is a very important research topic and the results of the study are also interesting enough to show the extreme sensitive to the scales at small time and space scales. As such, this paper have a big potential to be constructive to this topic. However, the results are presented alone without much explanation and details, also without comparing with some recent similar studies in details. Also the study does not dig into the assumption of the method it uses (idealized equations): the idealized equation is very useful to show some key characters but need to be

verified by comparing with more realistic models. These are shown by my comments below. Therefore, I encourage the author to have a major revision to include very thick explanation/analysis of their results. Then this study will be quite useful to this topic.

* sec 4: "The power input is found to increase monotonically with shorter coarse-graining in time but not with fi̧ner coarse-graining in space (see fi̧gs. 1 2)."

Why does it increase monotonically with shorter timescale? This needs explanation

* Figure 2: when lambda=0, the solid line case without eddy killing, why does the power input not always increase monotonically with smaller space scale? Eq1 shows the wind stress should definitely increase with smaller space scales, but eq4 indicates the wind correlation with the ocean current at different scale may have different magnitude and sign. You should explore this very carefully, otherwise your results are not deep enough. I would suggest you perform wavelength-frequency co-spectrum analysis, which can show clearly the scale-dependency of power input, see Figure 3 of Partitioning Ocean Motions Into Balanced Motions and Internal Gravity Waves: A Modeling Study in Anticipation of Future Space Missions, Journal of Geophysical Research, 123, 8084–8105.

* sec 4: "We further consider the eddy-killing through the parameter r kill (l,$\tau$) defi̧ned in eq. (5). It is positive for all averaging scales and time-intervals (l,$\tau$) (not shown), that is "eddy killing" always reduces the power input (fi̧g. 2 and also a comparison of the blue isolines and black isolines in fi̧g. 1). This comes at no surprise as it says that the ocean velocity at the surface and at 15m depth are on average positively correlated. "

The results here are interesting. However, it lacks details and explanation. - There are many studies there about eddy killing; how is your results similar or different comparing with past studies on eddy killing? Any new findings in your study. - your analysis on this results is too little; you should try to explain why you get different order of magnitude of eddy killing in different case, which can be new finding; is it due to different velocity difference between ocean and atmosphere in different case? is it always so in certain

region or vary dramatically from year to year (time dependent)? You should dig enough to explain why the blue case in figure 2 is so significant while the black case is minor? Rather than only presenting your calculation result without explanation.

* you results is based on the highly idealized and parameterized equation 4, and you use ocean current at 15m, which is limited since different ocean area have different mixed layer depth. Thus, how good is your method? you should compare with more realistic method/results. e.g., you should compare with the wind input in state of art coupled ocean model to make people trust your results, although this topic is really important. You should also compare with previous studies, e.g. do you have new findings in contrast to this paper: Global estimates of the energy transfer from the wind to the ocean, with emphasis on near-inertial oscillations. Journal of Geophysical Research: Oceans, 124, 5723–5746.

* sec5: "It is interesting to note that the strong decline in the power-input, in the extensions of the western boundary currents, between a day and a week is well fitted by a P(5 o ,$\tau$) âĹÌ $\tau$ −1/2 law (not shown). The increase of energy input at 10 days from 0.5 o to 4o (see fig 2) is well fitted by a P(l,10 days) âĹÌ l 1/2 ."

You should try to explain why, if any.

* figure 2 shows extreme sensitivity of power input to the small scales, e.g. blue and red dashed curves. Will this converge? This suggests that smaller scales may be important. So will the resolution of submesoscale be crucial for the power input, similar to the fact that submesoscale is key to the the vertical motions/fluxes in the oceans, some discussion will be useful, e.g. see/cite: An Annual Cycle of Submesoscale Vertical Flow and Restratification in the Upper Ocean. Journal of Physical Oceanography, 49, 1439–1461. Wind-forced symmetric instability at a transient mid-ocean front. Geophysical Research Letters, 46, 11,281–11,291; Ocean submesoscales as a key component of the global heat budget. Nature Communications, 9, 775.

* sec 5: "This high frequency forcing might not be important for the geostrophic circu-

Interactive
comment

lation but might be important only for the mixed layer, as noted by Zhai et al. (2012). this is why I considered the power input to the mixed layer, rather than the geostrophic circulation, in the present work."

Do your ocean current include the Ekman current and wave velocity? why or why not.

* line 5: "The abundant energy at the eddy scale ( $\approx$ 100km) in the ocean is found to be mostly supplied by baroclinic and barotropic instability of the basin scale circulation and the extensions of the western boundary currents to the ocean interior."

Recent studies indicate the possibility that another kinetic energy source of ocean mesoscale is from inverse cascade of KE from small scale (submesoscale) to large scales. It would be helpful to remind the readers of this. See/cite, e.g., Ocean-Scale Interactions from Space. Earth and Space Science, 6, 795-817; Impact of oceanic-scale interactions on the seasonal modulation of ocean dynamics by the atmosphere. Nature Communications, 5(1), 5636;

Also, for this sentence itself, it is helpful to the reader if you cite some papers that show the energy source of ocean mesoscale from baroclinic and barotropic instability.

* The writing should be polished further: e.g. "The power input is maximum for a coarse-graining of 6 ofor KUE and around 3 ofor GSE. Meaning that the dynamics at smaller scales reduces the power-input"

* line 20: "Eddy-killing is not restricted to eddies, its quantification over a large range of scales in space and time is considered here."

If Eddy-killing is not restricted to eddies, then why it is called "eddy"-killing? Please explain.

* line 30: "In the present work I determine the scales which are important for the mechanical power supply to the ocean. To this end a velocity vector is first averaged over a horizontal square of length l and a time interval $\tau$ to obtain the coarse-grained velocity"

Do you mean you first use mean velocity, then use higher-resolution velocity? you should explain it here. It is a little confusion.

---

## Referee Comment (RC2) · Anonymous Referee #2 · 8 Jun 2020

This is a potentially a very interesting paper which should be well read by the community. However, there is a major concern regarding the lack of detail in the discussion of your results and in particular the depth of reasoning in your paper. As it stands the paper will have to go through at least a major revision and possibly more experiments to strengthen your results presented here. I hope you will be encouraged to make these necessary improvements to your paper.

---

## Author Comment (AC2) · 30 Jun 2020

Dear Editor,

Please find below my detailed answer to the referee's comments. I am grateful to the referees for their work and also for qualifying my work as: " [… having] a big potential to be constructive to this topic." and
" […] a very interesting paper which should be well read by the community." Both referees criticism is centered around the discussion of the results " […] results are presented alone without much explanation and details" and " […] a lack of detail in the discussion". The discussion section is now completed along the referees' comments. But I like to add that the paper is based on observations and in this respect I want it to be empirical and invite the colleagues to come up with theoretical models, interpretations and comparisons, also based on modeling studies. Not comparing my observational data to model output is done on purpose, as model output should be compared to observational data, this is a small but important distinction. To my personal understanding:  in ocean sciences, it is the observational data and their literal analysis that is mostly lacking.

The referees are right when they ask for a theoretical explanation of the results. The results can, to my understanding, not be explained by back-of-the-enveloppe calculations but more involved approaches have to be used. I recently published two papers where analytical results on air-sea interaction were obtained in an idealized model using non-equilibrium statistical mechanics, the fluctuation-dissipation theorem (A. Wirth *J. Phys. Oceanogr.* (2018) **48** (4): 831–843  and A. Wirth, Nonlin. Processes Geophys., 26, 457–477, 2019 (elected NPG paper of the month)).  In this case eddy killing can be obtained analytically. Extending this work to consider space dependence means to consider the fluctuation-dissipation theorem of fields rather than particles. Such formalism has not been developed yet. Such formalism could potentially show the glass-transition of the air-sea system observed in Moulin, A., & Wirth, A. (2016).  *Boundary-Layer Meteorology*, *160*(3), 551-568. So we can obtain analytic results for idealized models and apply these results to less idealized models, but today the hierarchy of models  does not systematically connect the idealized models to observations. I have published papers on the idealized models and I wrote this paper on observations. Filling the gap in the hierarchy is a challenge for the future. Including analytical results in the present paper on observations could suggest that the gap is closed, but it is still widely open, to my understanding.

I added to the introduction:

Analytic results for local models of air-sea interaction, with and without eddy-killing, are given in \cite{wirth2019fluctuating}, where I also considered other implementations of air-sea interaction in coupled models.

I added a paragraph at the end of the manuscript:

The present work is empirical, no theory to explain the results is proposed In \cite{wirth2018fluctuation}  and \cite{wirth2019fluctuating}  the eddy-killing was obtained analytically for local models of air-sea-interaction. Extending this work to consider space dependence means to consider the fluctuation-dissipation theorem of fields rather than particles. Such formalism has not been developed yet. Constructing a hierarchy of models and approaches will further our understanding of air-sea interaction.

Sincerely,
Achim Wirth

The reviewers remarks are given in **blue**, my answers in **black** and the changes made in the manuscript in **red**.

Anonymous Referee #1

This study aims to explore the dependency of wind energy input to the ocean on the scales of time and space. This is a very important research topic and the results of the study are also interesting enough to show the extreme sensitive to the scales at small time and space scales. As such, this paper have a big potential to be constructive to this topic. However, the results are presented alone without much explanation and details, also without comparing with some recent similar studies in details. Also the study does not dig into the assumption of the method it uses (idealized equations): the idealized equation is very useful to show some key characters but need to be verified by comparing with more realistic models. These are shown by my comments below. Therefore, I encourage the author to have a major revision to include very thick explanation/analysis of their results. Then this study will be quite useful to this topic.

The present work is about anlyzing data. The power input is calculated using a simple but consistent model (see Wirth Nonlin. Processes Geophys., 26, 457–477, 2019 https://npg.copernicus.org/articles/26/457/2019/) A more involved model would include sea-surface roughness, but this data is not available for the same location and time and should be derived from other variables using models. This would add uncertainty to the my results.

* sec 4: "The power input is found to increase monotonically with shorter coarse-graining in time but not with finer coarse-graining in space (see figs. 1 & 2)."

Why does it increase monotonically with shorter timescale? This needs explanation

To my understanding, there is no law that imposes an increase with shorter time scales, it is an observation, a result that is found in my analysis and analysis performed by colleagues. This is explained in some detail in section 5 where the second and third paragraph are dedicated to this question.

The observation that the behavior is close to multiplicatively separable is a major result, that considerably reduces its complexity, not only when studying the problem but also when looking for parameterizations and so I added to the conclusion the sentence:

This also indicates that parameterizations for small-scales and short-times can be developed separately, reducing a two-dimensional problem to two one-dimensional problems.

I now added:

An increased power input due to higher frequency winds was also found in the numerical experiments of \cite{rimac2016total} and \cite{flexas2019global}. It points towards the fact that at short times there is a strong correlation between the ocean currents near the surface and the surface winds. Their data shows that inertial oscillations are responsible for the increase.

* Figure 2: when lambda=0, the solid line case without eddy killing, why does the power input not always increase monotonically with smaller space scale? Eq1 shows the wind stress should definitely increase with smaller space scales, but eq4 indicates the wind correlation with the ocean current at different scale may have different magnitude and sign. You should explore this very carefully, otherwise your results are not deep enough.

There is no result saying that eddy killing does not exist with lambda=0. In the results section it is and  was written:

"An "eddy-killing", that is a decrease of the energy input at small
scale appears also in the cases with $\lambda = 0$, (see blue contour-lines fig. 1 and also full lines fig. 2) it is however much less pronounced. The source of it may lie in the data of the wind velocity which might incorporate a degree of shear velocity, that is ocean dynamics."

I would suggest you perform wavelength-frequency co-spectrum analysis, which can show clearly the scale-dependency of power input, see Figure 3 of Partitioning Ocean Motions Into Balanced Motions and Internal Gravity Waves: A Modeling Study in Anticipation of Future Space Missions, Journal of Geophysical Research,123, 8084–8105.

The above cited publication does not talk or use co-spectrum analysis. The fig. 3 of the cited publication is  a  frequency-wave number spectra of KE. Performing such an analysis on the power-input determines the scales (time and space) at which the power enters the ocean. The question I answer using coarse graining is: what does the dynamics at a certain scale contribute to the power input. This is a different question (numerically much more involved).  On can apply a co-spectrum analysis as done in Flexas et al.,  (2019). Global estimates of the energy transfer from the wind to the ocean, with emphasis on near-inertial oscillations. JGR Oceans, 124, 5723–5746.
https://doi.org/10.1029/2018JC014453
(This paper was brought to my attention after the first submission, it is now cited)
But, please note also that the power input is not a multiplication of two dynamic variables (ocean-currents * atmospheric-winds, this is only the case when linear Rayleigh friction and lambda=0 is used) which would justify to use co-spectral analysis. The  real situation with a nonlinear dependence of the shear on atmospheric wind and oceanic currents, it is more involved (see my eqs. (1) and (2)). To my understanding co-spectral analysis is to simple to untangle the non-linear dynamics of air-sea interaction and coarse graining has to be used (which is numerically much more involved).

* sec 4: "We further consider the eddy-killing through the parameter r kill (l,τ) defined in eq. (5). It is positive for all averaging scales and time-intervals (l,τ) (not shown), that is "eddy killing" always reduces the power input (fig. 2 and also a comparison of the blue isolines and black isolines in fig. 1). This comes at no surprise as it says that the ocean velocity at the surface and at 15m depth are on average positively correlated. "The results here are interesting. However, it lacks details and explanation. - There are many studies there about eddy killing; how is your results similar or different comparing with past studies on eddy killing? Any new findings in your study. - your analysis on this results is too little; you should try to explain why you get different order of magnitude of eddy killing in different case, which can be new finding; is it due to different velocity difference between ocean and atmosphere in different case? is it always so in certain region or vary dramatically from year to year (time dependent)? You should dig enough to explain why the blue case in figure 2 is so significant while the black case is minor? Rather than only presenting your calculation result without explanation.

Even with the long data record used here it is difficult to obtain statistically significant results when restricted to seasons. When the eddy killing at one cut-off time-and-space-scale is considered my results are in agreement with already presented results. The originality of the present publication is (to my understanding) that I vary "continiously", not only one but both, the temporal and the spatial time-scale of the dynamic variables. I now added in the discussion section:

"The eddy killing is found to be especially strong in the extension of the western boundary currents, were the meso-scale and sub-meso-scale activity are strongest. In this regions the energy input can drop by a factor between two and three, due to eddy killing, when the averaging length scale is reduced from $5^o$ to $0.5^o$.

And also:

This agrees with the findings of \cite{renault2017satellite} who also found strongest eddy killing in the western boundary currents, but did not consider the scale dependence, but the effect of the atmospheric wind-speed. The scale dependence is found here to be a dominant effect in space and time.

My results show that scale dependence is essential. Finding analytic approximations for the power input had limited success in \cite{renault2017satellite} and I doubt that this is possible due to the scale dependence.

In the conclusion section I also added:

This agrees with a key point in \cite{flexas2019global}, who state that small scales in space and short times are critical to better representing wind power input in general circulation models.

I do not propose a parameterization of eddy killing as done in \cite{renault2017satellite}, because the scale dependence exposed in my research strongly suggests that the small scales in space and time should be explicitly resolved. This is also based on the discovery of a new instability mechanism when air-sea interaction is considered at fine resolution (Moulin, Aimie, and Achim Wirth. "A drag-induced barotropic instability in air–sea interaction." *Journal of physical oceanography* 44.2 (2014): 733-741.).

* you results is based on the highly idealized and parameterized equation 4, and you use ocean current at 15m, which is limited since different ocean area have different mixed layer depth. Thus, how good is your method? you should compare with more realistic method/results. e.g., you should compare with the wind input in state of art coupled ocean model to make people trust your results, although this topic is really important. You should also compare with previous studies, e.g. do you have new findings in contrast to this paper: Global estimates of the energy transfer from the wind to the ocean, with emphasis on near-inertial oscillations. Journal of GeophysicalResearch: Oceans, 124, 5723–5746.

This paper was brought to my attention after submitting the first version of my paper, it is now cited and discussed in the manuscript.
(Please see my discussion above and below on co-spectral analysis)

Eq. (4) is based on Newtons laws of physics and all parameterizations of air sea interaction are based on it. In the present paper I criticize simplifications of eq. 4 which are commonly used in ocean modeling (see also A. Wirth, Nonlin. Processes Geophys., 26, 457–477, 2019). The major unknown parameter is the drag coefficient, its determination is difficult because it depends on a variety of parameters, mostly the sea-surface roughness and also the temperature difference between the ocean surface and the atmosphere above. So to my understanding an ocean wave model should be key in realistic ocean simulations when fluxes of inertia are focused on. These models are highly non-linear and non-local in space-and time. Furthermore, my research and the research of colleagues clearly shows that coupled models have to be used, with a high resolution in the atmosphere that resolves the oceanic mesoscale and sub-mesoscale. The reviewer is right to point out the importance of the mixed layer depth and turbulence. These processes are somehow included in the velocity data at 15 depth, at least in a statistical sense (see (\cite{rio2014beyond} \cite{bonjean2002diagnostic} and \cite{sudre2013global} cited in the paper). In ocean models these processes are not explicitly resolved but parameterized based on empirical formulas. I am not sure that all these processes are well represented even in state of art coupled ocean models. And last but not least \cite{renault2017satellite} state:*"The wind response depends on the marine boundary layer height h: the shallower h, the larger a wind response, and, thus, the weaker sτ . However, no significant relationship between the mean h from ERA interim and sτ has been found."*

Near-inertial oscillations can not be considered with 6 hourly data due to aliasing error. Higher frequency data is not available.

This is a paper based on observations which should guide, be compared to ocean model outputs. Incorporating model outputs is conceptually difficult as it ultimately leads to ocean-model inter-comparison. I therefore refer the reader to a recent paper based on state of the art simulations.

I added to the introduction.

For a detailed discussion on the importance of scales in space-and-time for the power input to the ocean and the physical processes involved, I refer the reader to the recent work by \cite{flexas2019global}, who investigated the problem based on model results.

* sec5: "It is interesting to note that the strong decline in the power-input, in the extensions of the western boundary currents, between a day and a week is well fitted by a P(5 o ,τ) a τ^(−1/2) law (not shown). The increase of energy input at 10 days from 0.5 o to 4o (see fig 2) is well fitted by a P(l,10 days) l^(1/2) ."

You should try to explain why, if any.

The scaling of the power input towards shorter and faster scales is of key importance for parameterizations and has to the best of my knowledge not been mentioned elsewhere. I do not have an explanation for the exponents and I do not know how far the scaling goes. I currently investigate this question using a highly idealized QG-model introduced in Wirth, A. (2018). A fluctuation–dissipation relation for the ocean subject to turbulent atmospheric forcing. *Journal of Physical Oceanography*, *48*(4), 831-843.

* figure 2 shows extreme sensitivity of power input to the small scales, e.g. blue and red dashed curves. Will this converge? This suggests that smaller scales may be important. So will the resolution of submesoscale be crucial for the power input, similar to the fact that

submesoscale is key to the the vertical motions/fluxes in the oceans, some discussion will be useful, e.g. see/cite: An Annual Cycle of Submesoscale Vertical Flow and Restratification in the Upper Ocean. Journal of Physical Oceanography, 49, 1439–1461. Wind-forced symmetric instability at a transient mid-ocean front. GeophysicalResearch Letters, 46, 11,281–11,291; Ocean submesoscales as a key component of the global heat budget. Nature Communications, 9, 775.

I agree with the reviewer, smaller scales are important, but previous to having conducted the present research I would not have imagined to what point they are important for air-sea interaction. I wrote and write in the discussion:

"The power input further increases when even higher forcing frequencies are considered. This high frequency forcing might not be important for the geostrophic circulation but might be important only for the mixed layer, as noted by \cite{zhai2012wind}. this is why I considered the power-input to the mixed layer, rather than the geostrophic circulation, in the present work."

And also:

"The decay does not stop or saturate at the smallest scale considered. It means that air sea-interaction due to the sub-meso-scale dynamics substantially decreases the power-input."

In the conclusion I wrote and write and added the passage in red:

"The strongest gradients of power input are found for the smallest and shortest coarse-graining scales, that is within the coarse-graining domain considered here we do not see any convergence of the power input. The problem is further complicated by the opposing gradients in space (including smaller scales reduces the power input) and time (including shorter scales increases the power input). This indicates that higher resolutions in space and time are necessary to estimate the power fluxes into the ocean mixed layer and that we do not know at which scales we possibly observe convergence. This agrees with a key point in \cite{flexas2019global}, who state that small scales in space and short times are critical to better representing wind power input in general circulation models. Furthermore, we do not know to what extent the power input due to the dynamics at a certain scale in time and space contributes to climate relevant processes and how they can be parameterized. Our results also says that up- or down-scaling of data beyond the coarse-graining domain explored is not possible."

* sec 5: "This high frequency forcing might not be important for the geostrophic circulation but might be important only for the mixed layer, as noted by Zhai et al. (2012). this is why I considered the power input to the mixed layer, rather than the geostrophic circulation, in the present work."

Do your ocean current include the Ekman current and wave velocity? why or why not.

Yes, the Ekman velocity is calculated using an empirical model based on undrogued drifters and Argo floats. In the former a 3 day low-pass filter was applied to get rid of

inertial oscillations, stokes drift and other, while this was not possible for the Argo floats. I now added in the data section:

The ocean data at 15m depth is the sum of the geostrophic current  and the Ekman velocity at this depth. The latter was calculated using an empirical approach based on drifter data, as explained by \cite{rio2014beyond}.

* line 5: "The abundant energy at the eddy scale (≈100km) in the ocean is found to be mostly supplied by baroclinic and barotropic instability of the basin scale circulation and the extensions of the western boundary currents to the ocean interior."

Recent studies indicate the possibility that another kinetic energy source of ocean mesoscale is from inverse cascade of KE from small scale (submesoscale) to large scales. It would be helpful to remind the readers of this. See/cite, e.g., Ocean-Scale Interactions from Space. Earth and Space Science, 6, 795-817; Impact of oceanic-scale interactions on the seasonal modulation of ocean dynamics by the atmosphere. Nature Communications, 5(1), 5636;

Also, for this sentence itself, it is helpful to the reader if you cite some papers that show the energy source of ocean mesoscale from baroclinic and barotropic instability.

The process of baroclinc and barotropic instability is studied for the last 70 years and is present in all textbooks on the subject, that is why I referenced the book of Vallis which gives a nice and recent account of the subject. I changed to:

The energy is further distributed across scales by the non-linear interactions, possibly leading to energy cascades over extended ranges of scales, including  an inverse energy cascade from the submesoscale to larger scales (see  \cite{vallis2017atmospheric}, \cite{akuetevi2015dynamics}, \cite{klein2019ocean}).

* The writing should be polished further: e.g. "The power input is maximum for a coarse-graining of 6 o for KUE and around 3 o for GSE. Meaning that the dynamics at smaller scales reduces the power-input"

I now changed to:

The power input is maximum for a coarse-graining of $6^o$ for KUE and around $3^o$ for GSE. When smaller scales are included in the dynamics the power input is reduced.

* line 20: "Eddy-killing is not restricted to eddies, its quantification over a large range of scales in space and time is considered here."If Eddy-killing is not restricted to eddies, then why it is called "eddy"-killing? Please explain.

I know added:

Its impact was first noticed in some regions with strong eddy activity where it leads to a reduction of the energy at the eddy-scale of 30\% or more.

* line 30: "In the present work I determine the scales which are important for the mechanical power supply to the ocean. To this end a velocity vector is first averaged over a horizontal square of length l and a time interval τ to obtain the coarse-grained velocity" Do you mean you first use mean velocity, then use higher-resolution velocity? You should explain it here. It is a little confusion

The sentence is now changed to:

To this end a double average is taken: a velocity vector is first averaged over a horizontal square of length $l$ and a time interval $\tau$ to obtain the coarse-grained velocity:

There should be no ambiguity as the mathematical formula is presented right after.

I thank this referee for his thorough inspection of my work, that has considerably improved its quality.

Anonymous Referee #2

This is a potentially a very interesting paper which should be well read by the community. However, there is a major concern regarding the lack of detail in the discussion of your results and in particular the depth of reasoning in your paper. As it stands the paper will have to go through at least a major revision and possibly more experiments to strengthen your results presented here. I hope you will be encouraged to make these necessary improvements to your paper.

I thank the referee for finding my work "potentially a very interesting paper which should be well read by the community." I did perform changes suggested by the detailed comments of referee 1 and hope that this referee agrees with the changes. The paper presents a new and specific analysis of a very valuable data set. The results are new and surprising (to my understanding). I am convinced that my results are solid and can not come up with experiments to strengthen the point I make. The paper is empirical, adding simulations will weaken its statement. The explanation of the results present a challenge for the future.

---

## Author Comment (AC3) · 30 Jun 2020

Please find my answer to the referees in the pdf of the supplement

Please also note the supplement to this comment:
https://os.copernicus.org/preprints/os-2019-128/os-2019-128-AC3-supplement.pdf